# Recent Advances and Perspectives on the Use of Mineralocorticoid Receptor Antagonists for the Treatment of Hypertension and Chronic Kidney Disease: A Review

**DOI:** 10.3390/biomedicines13010053

**Published:** 2024-12-29

**Authors:** Kisho Miyasako, Yujiro Maeoka, Takao Masaki

**Affiliations:** Department of Nephrology, Hiroshima University Hospital, 1-2-3 Kasumi, Minami-ku, Hiroshima 734-8551, Japan; kisho.miyasako@gmail.com

**Keywords:** mineralocorticoid receptor antagonist, aldosterone, chronic kidney disease, diabetic kidney disease, diabetes, cardiovascular disease, finerenone, hyperkalemia, hypertension, epithelial sodium ion channel

## Abstract

Chronic kidney disease (CKD) is a major public health concern around the world. It is a significant risk factor for cardiovascular disease (CVD), and, as it progresses, the risk of cardiovascular events increases. Furthermore, end-stage kidney disease severely affects life expectancy and quality of life. Type 2 diabetes and hypertension are not only primary causes of CKD but also independent risk factors for CVD, which underscores the importance of effective treatment strategies for these conditions. The current therapies, including angiotensin-converting enzyme inhibitors, angiotensin II receptor blockers, and sodium–glucose co-transporter 2 inhibitors, are administered to control hypertension, slow the progression of CKD, and reduce cardiovascular risk. However, their efficacy remains suboptimal in certain instances. Mineralocorticoid receptor (MR), a nuclear receptor found in various tissues, such as the kidney and heart, plays a pivotal role in the progression of CKD. Overactivation of MR triggers inflammation and fibrosis, which exacerbates kidney damage and accelerates disease progression. MR antagonists (MRAs) have substantial beneficial effects in patients with cardiac and renal conditions; however, their use has been constrained because of adverse effects, such as hyperkalemia and kidney dysfunction. Recently, novel non-steroidal MRAs are more efficacious and have superior safety profiles to steroidal MRAs, making them promising potential components of future treatment strategies. Here, we discuss recent findings and the roles of MRAs in the management of hypertension and CKD, with a focus on the evidence obtained from fundamental research and major clinical trials.

## 1. Introduction

Mineralocorticoid receptor (MR) is a ligand-activated nuclear transcription factor that is expressed in various organs, including the kidney, heart, central nervous system, and vascular system [1]. MR binds to cortisol and aldosterone with similar affinity in vitro, although the primary ligands and functions of MR in vivo vary according to the target organ [2]. Physiologically, it is well-established that the aldosterone–MR pathway increases epithelial Na^+^ channel (ENaC) activity along the distal nephron, leading to increases in sodium and fluid retention and greater potassium excretion [3].

MR inhibition has antihypertensive effects by directly reducing ENaC activity and indirectly suppressing NaCl cotransporter (NCC) activity in the distal convoluted tubule (DCT), a response to a high serum K^+^ concentration [4,5]. Therefore, the administration of mineralocorticoid receptor antagonists (MRAs) is recommended in several guidelines, including the American Heart Association (AHA) guidelines, for patients with resistant hypertension (RHTN), which is a severe form of this condition [6]. Although MRAs are also extensively used in the management of chronic heart failure with reduced ejection fraction (HFrEF) to lower mortality rate and the risk of hospitalization because of heart failure [7], this benefit is considered to primarily stem from the cardioprotective effects of MR inhibition within the cardiovascular system rather than their antihypertensive effects. Indeed, MR is expressed not only in epithelial cells but also in non-epithelial cells, including vascular smooth muscle cells, podocytes, mesangial cells, and fibroblasts in the kidney. Activation of MR in these non-epithelial cells contributes to vascular dysfunction, glomerular damage, inflammation, and fibrosis. Consequently, MRAs would be expected to have beneficial effects beyond a reduction in blood pressure (BP), such as reductions in cardiac, glomerular, renal tubular, and interstitial tissue pathology, through their direct effects on various non-epithelial cells [8].

Non-steroidal MRAs, including esaxerenone and finerenone, have different pharmacological profiles to those of conventional steroidal MRAs (spironolactone and eplerenone) since they have different structures and bind differently within the MR ligand-binding domain. They show high affinity and selectivity for MR, and their use is associated with an extremely low incidence of sex hormone-related side effects, such as gynecomastia, and no pharmacologically active metabolites are found [8,9] (Figure 1). A robust body of evidence supporting the clinical utility of non-steroidal MRAs has now accumulated, particularly with respect to patients with hypertension [10,11] or chronic kidney disease (CKD) with type 2 diabetes (T2DM) [12,13,14]. In this review, we aim to summarize the recent clinical and basic evidence regarding the use of MRAs in patients with hypertension and CKD.

## 2. Methods

For this review, a PubMed search in “Title/Abstract” for “mineralocorticoid receptor antagonist” OR “mineralocorticoid receptor blocker” OR “MRA” AND “kidney” was performed until 30 November 2024, with no start date. After this search, 1065 results, editorials, comments on articles, conference abstracts, duplicates, and items unrelated to the research topic were removed. The remaining items were examined for content, and those that were consistent with the topic of this review were included as references. Articles authored by one or more co-authors of this review were independently assessed for validity by the other co-authors.

## 3. Efficacy of MRAs in Patients with RHTN

Hypertension is the most common chronic disease and the leading risk factor for cardiovascular disease (CVD) and premature death globally. Over one billion people were estimated to have hypertension worldwide in 2015, which contributed to 423 million cases of CVD and 18 million cardiovascular deaths annually [16]. This prevalence is expected to rise, owing to factors such as the aging of populations, unhealthy lifestyles, and the growing burden of comorbidities such as obesity and diabetes. RHTN is a severe form of hypertension that is characterized by poor BP control, despite treatment with at least three antihypertensive drugs, and poses a significant challenge because of its poor prognosis [17]. RHTN affects an estimated 10% of patients being treated for hypertension and therefore may affect over 100 million people globally. Evidence obtained from a recent large-scale analysis of randomized trials highlights the critical importance of BP control, showing that a 5-mmHg reduction in systolic BP is associated with a 10% reduction in the risk of major cardiovascular events, regardless of prior cardiovascular disease status [18].

The administration of MRAs is recommended in several guidelines, including the AHA guidelines, for patients with RHTN [6], as well as for those with cardiovascular, kidney, or liver disease. This recommendation has been supported by the results of several clinical randomized clinical trials [19]. Among these, the PATHWAY-2 (Prevention and Treatment of Hypertension With Algorithm-Based Therapy 2) trial, a double-blind, placebo-controlled, crossover trial of patients with RHTN who were already being treated with an angiotensin-converting enzyme inhibitor (ACEi) or an angiotensin receptor blocker (ARB), a calcium channel blocker, and a thiazide-like diuretic, demonstrated that the mean reduction in home-measured systolic BP achieved using spironolactone, an MRA, was superior to that achieved using the other treatments (MRA, −8.70 mmHg, 95% CI −9.72 to −7.69, *p* < 0.0001; α-blockers, −4.03 mmHg, 95% CI −5.04 to −3.02, *p* < 0.0001; β-blockers, −4.48 mmHg, 95% CI −5.50 to −3.46, *p* < 0.0001), indicating that an MRA is the most effective add-on treatment for RHTN [20]. More recently, Tian et al. performed a meta-analysis of 24 clinical studies that met their inclusion criteria, including a total of 3,458 patients, in which they analyzed the effects of 12 pharmacological, surgical, and non-surgical treatments on the BP of patients with RHTN. Of these, spironolactone was the most effective at reducing home-measured BP in patients with RHTN (−8.46 mmHg, 95% CI −12.54 to −4.38, *p* < 0.0001) [21].

There is also mounting evidence of a cardioprotective effect of MRAs in patients with RHTN. Tsujimoto et al. studied patients who had heart failure with preserved ejection fraction (HFpEF) with and without RHTN in the TOPCAT (Treatment of Preserved Cardiac Function Heart Failure With an Aldosterone Antagonist) trial [22]. They showed that in those with HFpEF and RHTN, spironolactone significantly reduced the risk of cardiovascular events versus placebo (hazard ratio (HR) 0.70, 95% CI 0.53–0.91, *p* = 0.009), while no significant effect was observed in those without RHTN (HR 1.00, 95% CI 0.83–1.20, *p* = 0.97). Although spironolactone reduced the BPs of patients both with and without RHTN, a larger reduction in BP was achieved in those with HFpEF and RHTN (−4.4 versus −1.8 mmHg, *p* = 0.006) [22], suggesting that the effective control of BP might reduce the risk of cardiovascular events in patients with RHTN. Similarly, Krasińska et al. found that the addition of eplerenone to standard antihypertensive therapy significantly ameliorated left ventricular hypertrophy (LVH) and reduced night-time BP in patients with obstructive sleep apnea and RHTN [23]. In addition, BP-independent effects of MRAs on cardiac function have been identified in patients with RHTN. Spironolactone treatment of such patients has been shown to significantly reduce left atrial size and improve atrial function and atrioventricular coupling, independently of their baseline aldosterone concentrations [24]. In addition, it was shown to improve the properties of the aorta, such as by decreasing pulse-wave velocity and increasing pulsatility and distensibility [25]. Therefore, MRAs may reduce the risk of cardiovascular events and ameliorate LVH in patients with RHTN, primarily through reducing BP but potentially also through direct cardioprotective effects.

## 4. Efficacy of MRA in Patients with Low-Renin Hypertension

Low-renin hypertension (LRH), commonly defined as a plasma renin activity of <1 ng/mL/hour [26], is present in 30% of patients with hypertension [27,28]. Studies of patients with RHTN have shown that in those with low renin concentrations, MRAs are more effective at reducing BP than α- or β-blockers. For example, at a plasma renin level of 1 mU/L, estimated reductions in systolic blood pressure are as follows: spironolactone (−25.2 mmHg), doxazosin (−11.6 mmHg), and bisoprolol (−9.3 mmHg), based on individual linear regression analyses depicted in the figures [6,29]. Therefore, it has been hypothesized that LRH may reflect excess MR activation, likely because of abnormalities in renal sodium handling in the distal nephron segments of the kidneys [30]. However, current clinical practice guidelines do not provide clear recommendations regarding the initial choice of an optimal approach to MRA therapy for people with LRH [31].

Recently, Shah et al. performed a meta-analysis of 17 clinical studies in which the efficacy of the MRA therapy of patients with LRH was assessed. They revealed that MRA reduced systolic BP by mean values of −6.83 mmHg (95% CI −9.56 to −4.1) and −4.75 mmHg (95% CI −11.91 to +2.4) versus the use of an ACEi or ARB, respectively, plus diuretics [32]. These findings suggest that MRAs are effective at reducing the BPs of patients with LRH and may be superior to ACEis/ARBs. In a retrospective single-center cohort study, Mansur et al. assessed the effects of MRA therapy on reducing the BP and/or proteinuria of patients with LRH or probable primary aldosteronism (PA). The study showed that an increase in renin concentration is a useful biomarker of reductions in BP and proteinuria. In addition, similar BP reductions were achieved, regardless of the aldosterone concentrations of the patients, which permitted most to discontinue their other antihypertensive medications [33]. These results suggest that MRAs may be more effective than ACEis/ARBs and diuretics in patients with LRH and that the treatment of unsuppressed renin concentrations using MRAs might be beneficial, as in patients with PA. As a means of potentially validating these findings, the results of the Randomized trial assessing the Efficacy and safety of Mineralocorticoid receptor Antagonist therapy compared to Standard antihypertensive Therapy in hypErtension with low Renin (REMASTER) are eagerly awaited [34].

## 5. Efficacy of Esaxerenone in Patients with Hypertension

Esaxerenone was the first non-steroidal MRA to be approved in Japan (in 2019) for the treatment of essential hypertension. However, it is not currently approved by the Food and Drug Administration (FDA) or the European Medicines Agency (EMA). It undergoes hepatic metabolism and has a long half-life (Figure 1) [8,9]. Mounting evidence shows that esaxerenone is more efficacious than spironolactone, eplerenone, trichlormethiazide, or other thiazide diuretics in Japanese patients with essential hypertension. Furthermore, the ESAX-HTN study, a phase III multicenter, randomized, double-blind study, demonstrated that esaxerenone has BP-lowering activity that is at least equivalent to that of eplerenone in Japanese patients with essential hypertension [35]. A post-hoc analysis of data from the ESAX-HTN study showed that the night-time systolic BP reductions were significantly larger in patients administered esaxerenone at 2.5 or 5 mg per day than in that administered eplerenone (eplerenone, −2.6 mmHg, 95% CI −5.0 to −0.2 mmHg; esaxerenone, −6.4 mmHg, 95% CI −8.8 to −4.0). The analysis also showed that the esaxerenone-induced reductions in night-time BP were more pronounced in older patients, suggesting that esaxerenone may be particularly effective for the management of nocturnal hypertension in older patients and in those with a “non-dipper” BP pattern [11]. The EXCITE-HT study, a multicenter, randomized, open-label, parallel-group study of Japanese patients with uncontrolled essential hypertension who were already on ARB or calcium channel blocker treatment, showed that both esaxerenone and trichlormethiazide reduced BP. However, esaxerenone administration was associated with lower incidences of changes in serum K^+^ concentration and uric acid concentration than trichlormethiazide administration, suggesting that esaxerenone is safer with respect to electrolyte balance [10]. Recently, the efficacy and safety of esaxerenone were evaluated in patients with moderate kidney dysfunction. Two multicenter, open-label, nonrandomized dose-escalation studies were conducted in which the administration of esaxerenone as a monotherapy and as an add-on to renin-angiotensin system inhibitor therapy was compared in Japanese patients with hypertension and moderate kidney dysfunction. These studies demonstrated that esaxerenone reduces the BPs of such patients when administered in either fashion (monotherapy, −18.5/−8.8 (95% CI −23.7 to −13.3/−11.9 to −5.7) mmHg, *p* < 0.001; add-on therapy, −17.8/−8.1 (95% CI −21.0 to −14.7/−9.7 to −6.5) mmHg, *p* < 0.001) [36]. These results suggest that esaxerenone is an effective and well-tolerated therapy, even in patients with moderate kidney dysfunction.

## 6. Mechanisms Whereby MR Causes Aldosterone-Dependent and Independent ENaC Activation

The PATHWAY-2 mechanism sub-studies revealed that spironolactone reduces BP by reducing thoracic volume, which highlights the importance of its natriuretic effect [29]. They also showed that the BP response is only weakly predicted by the plasma aldosterone concentration [29]. Similarly, sub-analyses of data from the RENALDO study, a double-blind, randomized, crossover-controlled trial, revealed that many patients with low or normal aldosterone concentrations still respond to MRAs [37]. Therefore, although screening patients with RHTN for PA remains essential, the administration of MRAs is often recommended, even when the patients’ aldosterone concentrations are not high [6], raising the possibility that some MRs are activated by hormones other than aldosterone. Given that glucocorticoid hormones can also activate MR and typically circulate at concentrations 100–1,000 times higher than those of aldosterone, which results in MR activation in most cells, aldosterone-independent MR activation, possibly mediated by glucocorticoids, may be critical for sodium homeostasis.

Recently, we investigated the aldosterone-independent activation of the MR–ENaC pathway by comparing the nuclear MR expression, apical α-ENaC expression (indicating ENaC activation), and phenotype of kidney-specific MR-deficient (MR-KO) mice and aldosterone synthase-deficient (AS-KO) mice on a C57BL/6 background [38]. The MR-KO mice showed low nuclear MR and apical α-ENaC expression in both the late DCT (DCT2)/the early connecting tubule (CNT1) and the late connecting tubule (CNT2)/cortical collecting duct (CCD) (Figure 2). By contrast, AS-KO mice showed low MR and α-ENaC expression in the CNT2/CCD, and normal expression in the DCT2/CNT1 (Figure 2). Phenotypically, the MR-KO mice exhibited a salt-losing state, high serum K^+^ concentrations, and low serum Na^+^ concentrations, whereas the AS-KO mice exhibited only slightly high serum K^+^ concentrations and slightly low phosphorylated NCC (pNCC) expression. Notably, whereas MR-KO mice showed persistently high Na^+^ excretion when consuming a low-salt (LS) diet, in AS-KO mice the high Na^+^ excretion was only temporary. However, eplerenone administration to AS-KO mice consuming an LS diet induced a salt-losing state that was similar to the phenotype of the MR-KO mice, characterized by a high level of Na^+^ excretion, high serum K^+^ concentration, and high blood urea nitrogen concentration [38] (Figure 2), suggesting that the MR–ENaC pathway is important for the role of DCT2/CNT1 in sodium homeostasis. This contention is supported by the results of experiments using several animal models and patch clamps. Furthermore, mice in which γ-ENaC and MR were deleted in the epithelial cells along the entire nephron exhibited salt-wasting syndrome and hyperkalemia [4,39,40], whereas those with a deletion of MR along CNT2 and CCD exhibited only a mild phenotype, characterized by normal renal Na^+^ excretion but a high aldosterone concentration [41]. Patch clamp experiments demonstrated that the ENaC current is larger along DCT2/CNT1 than CNT2/CCD [42,43]. These findings suggest that the activation of MR–ENaC in DCT2/CNT1 occurs independently of aldosterone and plays a crucial role in sodium reabsorption, consistent with the notion that MRAs have potent antihypertensive effects, even in patients with RHTN and low-to-normal plasma aldosterone concentrations. In addition, whereas MR was highly expressed in DCT2, the expression of 11β-hydroxysteroid dehydrogenase type 2, which inactivates cortisol, was very low [38], consistent with the activation of MR in DCT2/CNT1 by cortisol (Figure 2).

## 7. Definition and Severity Classification of Chronic Kidney Disease (CKD)

CKD has recently been recognized as a significant global public health problem [44]. In 2017, approximately 700 million people worldwide were estimated to have CKD, corresponding to a prevalence of 9.3% [44]. This prevalence is predicted to increase because of factors such as the aging of populations, the increases in the number of people with T2DM and hypertension [45,46], the low early diagnosis rate of CKD, and treatment inertia [47]. Despite substantial healthcare investments, the life expectancy of patients with CKD remains significantly shorter than that of the general population [48]. This shorter life expectancy is attributable to the progression of CKD and related complications, such as end-stage kidney disease (ESKD) and CVD [49]. Therefore, addressing these risks and improving the outcomes of patients with CKD are critical global health priorities [50].

CKD is defined as the presence of a structural or functional abnormality of the kidney that persists for more than 3 months, accompanied by either a low glomerular filtration rate (GFR) or the presence of markers of renal impairment [51]. These structural abnormalities include abnormalities on histology or imaging and a history of kidney transplantation, and the functional abnormalities include albuminuria, defined using an albumin-to-creatinine ratio (ACR) ≥30 mg per gram creatinine (approximately ≥3 mg/mmol), an albumin excretion of ≥30 mg/day, a GFR < 60 mL/min/1.73 m^2^, abnormalities in the urine sediment, and electrolyte or other abnormalities caused by tubular disorders. The presence of one or more of these is essential for a diagnosis to be made. In particular, an increase in albuminuria or proteinuria is a well-established potent risk factor for both ESKD and CVD in patients with CKD, irrespective of their diabetes status. A high level of albuminuria is independently associated with a high risk of death, irrespective of the estimated GFR (eGFR) [51]. The CKD classification developed by Kidney Disease Improving Global Outcomes (KDIGO), which is determined using the level of albuminuria, GFR, and disease etiology, provides a means of estimating overall patient risk. This classification closely correlates with the risk of death, CVD, and progression to ESKD [51]. Therefore, in clinical practice, patients who are identified to be at high risk on the KDIGO heat map should undergo intensive prophylactic treatment to mitigate the risk of developing CVD and the progression to ESKD.

## 8. Efficacy of MRAs in Patients with CKD and Type 2 Diabetes

In patients with CKD and T2DM, renin–angiotensin–aldosterone system inhibitor (RAAS-i) therapy is well-established as the principal means of slowing CKD progression [52,53]. However, combination therapy with multiple RAAS-is does not improve cardiovascular or renal outcomes, but rather it is associated with higher risks of hyperkalemia, hypotension, and acute kidney injury (AKI) [54,55]. Spironolactone and eplerenone, which are efficacious in patients with HFrEF in the RALES and EMPHASIS-HF trials, have also been shown to reduce albuminuria in patients who are already undergoing RAAS-i therapy [15,56], indicating that they may also be beneficial for patients with CKD. However, the limited evidence regarding long-term renal outcomes, the risk of hyperkalemia owing to residual renal dysfunction, and the challenges of using eplerenone in patients with severe renal impairment have restricted its widespread use in such patients.

Recently, two phase III trials performed in patients with CKD, T2DM, and serum K^+^ concentrations <4.8 mmol/L who are undergoing therapy with maximal doses of RAAS-i have demonstrated that the non-steroidal MRA finerenone reduces the risk of cardiovascular events and inhibits CKD progression [12,13] (Figure 3A). These two studies are Finerenone in Reducing Kidney Failure and Disease Progression in Diabetic Kidney Disease (FIDELIO-DKD) [12] and Finerenone in Reducing CV Mortality and Morbidity in Diabetic Kidney Disease (FIGARO-DKD) [13]. In the FIDELIO-DKD study, finerenone significantly reduced the risks of renal failure, a sustained decline in eGFR from baseline of ≥40%, or death from renal causes versus placebo (HR 0.82, 95% CI 0.73–0.93). In the finerenone group, the cardiovascular outcomes were also improved, with lower risks of cardiovascular death, nonfatal myocardial infarction, nonfatal stroke, or hospitalization because of heart failure. The renoprotective and cardioprotective effects of finerenone were also confirmed in the FIGARO-DKD study. The FIDELITY analysis was an analysis of the data from both trials, which comprised 13,026 patients who were followed for a median of 3 years, and of whom 66.3% had CKD stages G1–G3b. The analysis showed significantly lower incidences of events of a composite cardiovascular endpoint (time to first onset of cardiovascular death, nonfatal myocardial infarction, nonfatal stroke, and hospitalization because of heart failure; HR 0.86, 95% CI 0.78–0.95, *p* = 0.0018) and hospitalization because of heart failure (HR 0.78, 95% CI 0.66–0.92, *p* = 0.0030) [57] (Figure 3B). Similarly, finerenone reduced the time-to-event risk of a composite renal endpoint (onset of renal failure, a sustained decrease in eGFR of ≥57%, lasting >4 weeks from baseline, or death from a renal cause; HR 0.77, 95% CI 0.67–0.88, *p* = 0.0002) and delayed the initiation of renal replacement therapy (maintenance dialysis or renal transplantation; HR 0.80, 95% CI 0.64–0.99, *p* = 0.040) (Figure 3C). The consistency in the results obtained regarding all the renal endpoints reinforces the results of the FIDELIO-DKD and FIGARO-DKD trials. Notably, the systolic BP of patients was reduced by only 2.1 mmHg in the finerenone group (+0.9 mmHg in the placebo group), implying that the cardiovascular and renal benefits of this drug are exerted largely independently of its antihypertensive effect [58]. On the basis of these results, the KDIGO consensus meeting recommended that finerenone should be administered to patients with eGFR ≥25 mL/min/1.73 m^2^, normal serum K^+^ concentrations, and ACR ≥ 30 mg/gCr who are being administered the maximum dose of RAAS-is [59].

As for finerenone, esaxerenone was developed to improve the efficacy and safety of therapy in particular while reducing the hyperkalemia associated with steroidal MRA use. The ESAX-DN trial demonstrated that esaxerenone, like finerenone and other MRAs, reduces albuminuria but also has a more potent hypotensive effect (systolic BP: −10 mmHg) [14]. This ESAX-DN study was conducted in Japanese patients aged ≥20 years who had both hypertension and T2DM. The patients were also treated with an RAAS-i for at least 12 weeks, had a urinary ACR of 45 to 300 mg/gCr, and had an eGFR ≥ 30 mL/min per 1.73 m^2^. In the final analysis of 449 patients after 52 weeks of follow-up, the proportion who achieved the primary endpoint of urinary ACR remission (defined as an ACR of <30 mg/gCr and a ≥30% reduction from baseline on two consecutive occasions) was higher in the esaxerenone group than in the placebo group (22% versus 4%; absolute difference 18%, 95% CI 12–25%, *p* < 0.001). Although the rate of change in urinary ACR from baseline to the end of treatment was significantly higher in the esaxerenone group than in the placebo group (−58% versus 8%, geometric least-squares mean ratio versus placebo 0.38, 95% CI 0.33–0.44), hyperkalemia was more frequent in the esaxerenone group, necessitating dose reductions or discontinuation in some patients. In addition, there was no close correlation between the change in urinary ACR with the reduction in BP or eGFR. Thus, esaxerenone, like finerenone, appears to be an effective treatment option for patients with CKD and T2DM because of its significant renoprotective effects, including a reduction in albuminuria. However, several challenges and gaps in the evidence base remain. The higher incidence of hyperkalemia associated with this treatment necessitates careful monitoring and potential dose adjustments. Furthermore, ethnicity-related differences and long-term renal outcomes have not been fully explored, and therefore further research is necessary to validate its efficacy and safety in a range of populations.

## 9. Strategies to Manage the Adverse Effects of MRAs in Patients with CKD

Patients who have advanced CKD are often administered ARBs or ACEis and typically exhibit poor residual renal function. Consequently, the effects of MRAs on hyperkalemia and renal function are a critical clinical concern [60]. MRAs have been reported to significantly increase the incidence of hyperkalemia in patients with CKD stages G1–G4 [61,62]. In addition, a meta-analysis showed that the combination of an ARB or ACEi with an MRA significantly increases the risk of hyperkalemia [62].

In the two principal trials of finerenone (FIDELIO-DKD and FIGARO-DKD) described above, the patients’ serum K^+^ concentrations were managed according to a predefined algorithm [63] (Figure 4A). Although the incidence of hyperkalemia was higher in the finerenone group, only 1.7% of the patients in the finerenone group discontinued their medication (0.6% in the placebo group), a very low incidence of severe hyperkalemia and no resulting deaths were reported [57] (Figure 4B). Despite the low eGFR values of the participants, the proportion who discontinued finerenone was lower than that for other MRAs (Figure 4B). This result suggests that finerenone has a smaller effect on serum K^+^ concentration, making it safer for patients with poor renal function [14,15,56]. The lower overall incidence of hypokalemia in patients taking finerenone is likely the result of the equal distribution of finerenone to the heart and kidneys, in contrast to steroidal MRAs, which primarily concentrate in the kidneys, as well as its shorter half-life and the absence of active metabolites [64,65]. The use of the hyperkalemia management algorithm from these trials could reduce the incidence of adverse events.

A post-hoc analysis of data from the FIDELIO-DKD trial revealed that the risk of hyperkalemia is higher in patients with baseline serum K^+^ concentrations >4.5 mmol/L, eGFR < 45 mL/min/1.73 m^2^, and high ACR, and in those taking β-blockers; and that it is higher when used alongside diuretics and sodium–glucose co-transporter 2 inhibitors (SGLT2is) [63]. The effect of SGLT2is to reduce the risk of hyperkalemia has been previously demonstrated when they have been administered in combination with eplerenone, esaxerenone, or finerenone [66,67,68], suggesting that the use of a combination of an SGLT2i and diuretics may play an important role in the management of hyperkalemia. In addition, the management of MRA-induced hyperkalemia using potassium adsorbents is under investigation. The AMBER trial, a phase II randomized, double-blind, placebo-controlled, multicenter study conducted across 10 countries, including Europe, the United Kingdom, South Africa, and the United States [69], investigated this approach. Approximately 300 patients were enrolled (half female, 99% White) who had refractory hypertension and comorbid CKD, characterized by eGFRs of 25–45 mL/min/1.73 m^2^ (mean eGFR 35 mL/min/1.73 m^2^) and serum K^+^ concentrations of 4.3–5.1 mmol/L. Of the participants, approximately 50% were taking β-blockers, 70% were taking calcium channel blockers, and almost all were taking diuretics, along with ACEis or ARBs. The eligible patients were randomly allocated to either a patiromer or placebo group. The primary endpoint, the continuation of spironolactone treatment after 12 weeks, was achieved significantly more frequently in the patiromer group (86% versus 66%, 95% CI 10.0–29.0, *p* < 0.0001), and more patients were able to continue taking a spironolactone dose of 50 mg/day than placebo (69% versus 51%, but with no significant difference). As expected, hyperkalemia (serum [K^+^] > 5.5 mmol/L) was more prevalent in the placebo group (approximately 65% versus 35% was inferred, although no significant difference was described) and explained nearly two-thirds of the instances of discontinuation of spironolactone administration. These findings suggest that potassium adsorbent administration may enable the safe administration of ACEis or ARBs with MRAs to patients at a high risk of developing hyperkalemia.

Studies of the effects of MRAs on renal function have generated contradictory findings, with some showing no effect [70] and others a worsening of renal function [71]. In a large randomized controlled trial (RCT) of patients with heart failure and/or left ventricular dysfunction, steroidal MRA therapy was associated with a decrease in eGFR and a concomitant increase in serum creatinine concentration immediately after its initiation [72]. However, recent findings suggest that the high serum creatinine concentrations that develop following the initiation of an MRA are likely the result of renal hemodynamic effects and may not adversely affect prognosis [73]. This phenomenon, known as “pseudo-worsening” renal function in patients with advanced CKD, may contribute to the challenge associated with the maintenance of MRA therapy. Indeed, it has been reported that patients who discontinue MRA administration, especially those with low eGFR, are less likely to have therapy reinstated [74]. Despite these changes in the creatinine and K^+^ concentrations, the beneficial effects of MRA therapy have been maintained in clinical trials, and the initial decline in renal function may be similar to the initial reduction in eGFR that is associated with RAAS-i and SGLT2i use [75,76].

Taking these findings together, the use of MRAs in patients with CKD should be carefully considered, weighing the benefits and potential risks. Notably, adverse effects such as hyperkalemia may be alleviated by the concomitant administration of diuretics, SGLT2is, or potassium adsorbents. Future treatment strategies should be guided by robust evidence as it accumulates.

## 10. Efficacy of MRAs for the Treatment of Advanced CKD

Several studies have shown that patients with advanced CKD are at a high risk of cardiovascular events. 50% of patients with CKD stages G4–G5 have CVD [77], and whereas cardiovascular deaths in patients with normal renal function account for 26% of all deaths, in patients with CKD stages 4–5 they account for approximately 40–50% [78,79].

The administration of esaxerenone or eplerenone is generally not recommended for patients with CKD stages G4 or G5, whereas finerenone or spironolactone can be administered cautiously to those who are not anuric or in acute renal failure. A pooled analysis of RCTs of steroidal MRAs was published in 2022 that included a total of 12,700 patients, 331 (2.6%) of whom had CKD stage G4 or G5 and an eGFR ≤ 30 mL/min/1.73 m^2^ [80]. In this study, steroidal MRA administration reduced the risk of a combination of cardiovascular death or hospitalization because of heart failure versus placebo, although the effect diminished as eGFR decreased. Notably, there was no significant risk reduction associated with MRA treatment in patients with eGFR ≤ 30 mL/min/1.73 m^2^ (HR 0.96, 95% CI 0.70–1.3), and the risks of hyperkalemia (HR 2.09, 95% CI 1.38–3.1) and a deterioration of kidney function (HR 2.05, 95% CI 1.24–3.41) were approximately two-fold higher. Therefore, when treating such patients, the relative lack of efficacy of steroidal MRAs and their associated adverse effects, such as hyperkalemia and renal dysfunction, are critical considerations. These findings are consistent with the results of cohort studies performed in Alabama and Taiwan, which showed a similar lack of efficacy in patients with advanced CKD [81,82,83]. However, the FIDELITY analysis showed the benefits of finerenone, even in patients with eGFR < 45 mL/min/1.73 m^2^ (CKD stage G3b–G4, 33.7% of the total). The analysis suggested that finerenone reduces the risk of hospitalization because of heart failure for patients with CKD stages G3b–G4, as well as in those with stages G1–G3b [57].

In conclusion, steroidal MRAs are effective in patients with CKD stages G1–G3b, and finerenone is effective in those with CKD stages G1–G4. However, close monitoring for hyperkalemia and worsening kidney dysfunction is necessary. Larger, dedicated trials of the administration of both steroidal and non-steroidal MRAs to patients with advanced CKD are needed to corroborate these findings.

## 11. Mechanisms of MR Activation in Models of Diabetes

The inflammation and fibrosis caused by MR overactivation are major contributors to the progression of DKD. Studies of several animal models have demonstrated that MR activation induces oxidative stress in the kidney [84] and is a key mediator of renal inflammation and fibrosis [85]. In addition, MR activation is driven by factors such as a low GFR and a high aldosterone concentration secondary to renal dysfunction, as well as hyperglycemia and insulin resistance [86], suggesting that patients with DKD are likely to have a highly active degree of MR activation. Furthermore, the suppression of MR overactivation ameliorates renal and vascular inflammation and mitigates myocardial damage [87,88]. Therefore, targeting MR to reduce inflammation and fibrosis, which are key drivers of the progression of DKD, may be an effective therapeutic approach.

RAS-related C3 botulinum toxin substrate 1 (Rac1), a low-molecular-weight G protein, enhances the transcriptional activity of MR, independently of its ligand [89]. Activation of the Rac1–MR pathway is associated with renal abnormalities, including severe albuminuria and podocyte damage [89]. Recently, Hirohama et al. investigated the role of the Rac1–MR pathway in type 2 diabetes using *db*/*db* mice fed a high-salt (HS) diet for 10 weeks after unilateral nephrectomy (UNx) (UNx-HS *db*/*db*) [90]. Compared to the control group (UNx-HS *db*/*m*), the UNx-HS *db*/*db* mice showed hypertension, accompanied by massive albuminuria, glomerular damage with nodular lesions, and high expression of serum- and glucocorticoid-induced protein kinase 1 (SGK1), a target of MR, and truncated ENaC. In addition, the expression of the active form of GTP-Rac1 was high in glomerular podocytes. These effects were ameliorated by treatment with finerenone or NSC23766, a Rac1 inhibitor. These findings indicate that activation of the Rac1–MR pathway, which is induced by HS intake, contributes to glomerular damage and hypertension in T2DM, particularly through its effects on podocytes and distal nephrons (Figure 5A) [90].

O-linked-N-acetylglucosamine (O-GlcNAc) modifications of nuclear and cytoplasmic proteins play a crucial role in glucose sensing. Jo et al. demonstrated that high glucose concentrations or certain drugs promote O-GlcNAcylation by glucosamine and O-(2-Acetamido-2-deoxy-D-glucopyranosylidene)amino N-phenyl carbamate in the presence of aldosterone-enhanced MR transcriptional activity and SGK1 expression [91]. They also found that MR undergoes O-GlcNAc modification at Ser295–Ser307 and that mutations at Ser295, Ser298, and Ser299 reduce O-GlcNAcylation, MR protein levels, and transcriptional activity using liquid chromatography–tandem mass spectrometry. In a model of type 2 diabetes (*db*/*db* mice), MR protein, *Sgk1* mRNA, and O-GlcNAc protein expression levels were found to be high in the kidney. Furthermore, treatment with an inhibitor of O-GlcNAcylation reduced the O-GlcNAc protein levels, leading to a subsequent decrease in MR protein expression (Figure 5B). Although the specific kidney cell types in which MR undergoes O-GlcNAcylation are unknown, it has been hypothesized that this modification stabilizes MR and enhances aldosterone sensitivity in individuals with T2DM [91].

Lyngsø et al. developed a model of type 1 diabetes by administering low-dose streptozotocin (STZ) to endothelial cell-specific MR knockout (ECMR-KO) and wild-type mice over 5 days [92]. Diabetic wild-type (STZ WT) mice had enlarged hearts and kidneys and showed substantial urinary albumin excretion, whereas diabetic ECMR-KO (STZ ECMR-KO) mice showed no enlargement of their hearts or kidneys and less urinary albumin excretion. In addition, the acetylcholine-dependent vasorelaxation response in mesenteric arteries was impaired in STZ WT mice, while it was preserved in STZ ECMR-KO mice. These findings indicate that MR activation in endothelial cells contributes to the endothelial dysfunction that characterizes diabetes, and they suggest that MR inhibition in endothelial cells may have protective effects on both the cardiovascular system and kidneys (Figure 5C) [92].

These results are consistent with the notion that MR activation in podocytes and endothelial cells is a key contributor to albuminuria and that MRAs reduce albuminuria through direct effects on these cells. Moreover, this mechanism may explain the reduction in albuminuria that is observed in patients with CKD complicated by T2DM who administer an RAAS-i, an effect that is independent of the antihypertensive effects of the MRAs [12,14,15,56,66].

## 12. Evidence for the Use of MRAs in Patients Undergoing Dialysis

To the best of our knowledge, the evidence for the use of MRAs in patients undergoing renal replacement therapy is almost exclusively related to the use of spironolactone and eplerenone. In 2019, Charytan et al. reported the results of the randomized, placebo-controlled, double-blind SPin-D study [93]. This was an early-stage, dose-range study that aimed to determine the safety and efficacy of spironolactone (12.5, 25, or 50 mg/day) in 129 patients who were undergoing chronic hemodialysis and were followed for up to 3 years. During this study, 27 patients (21%) completely discontinued their medication, primarily because of hyperkalemia, and evidence showed higher incidences of hyperkalemia and hypotension when the drug was administered at higher doses, particularly at 50 mg/day. However, there were no overall differences between the spironolactone and placebo groups with respect to the two primary safety outcomes of a K^+^ concentration > 6.5 mmol/L and hypotension requiring hospitalization or a visit to the emergency department. Although no clear improvements in cardiac function (diastolic function, assessed using echocardiography) or structure (left ventricular mass index, LVEF, or global left ventricular longitudinal strain) were identified, the authors concluded that spironolactone is well tolerated, by comparison to placebo, in patients undergoing maintenance hemodialysis. In addition, Hammer et al. reported the results of the MiREnDa study, which aimed to determine the effect of spironolactone 50 mg once daily versus placebo on the left ventricular mass (LVM) of 97 patients who had undergone chronic hemodialysis and were randomized 1:1 in a double-blind fashion [94]. They found that 79.4% of the patients who were randomized took at least half a tablet of the study drug daily for at least 30 weeks, equating to an overall mean dose of spironolactone of 42 mg/day. Treatment with spironolactone 50 mg/day had no significant effect on LVM, left-ventricular ejection fraction, or other cardiovascular end-points, including 24-h systolic or diastolic BP. Spironolactone 50 mg/day treatment was associated with a significantly higher incidence of moderate hyperkalemia (defined as a pre-dialysis K^+^ concentration of 6.0–6.5 mmol/L, spironolactone 155 events versus placebo 80 events, *p* = 0.034), whereas the incidence of severe hyperkalemia did not significantly differ between the groups (pre-dialysis K^+^ concentration ≥ 6.5 mmol/L, spironolactone 14 events versus placebo 24 events, *p* = 0.225). The principal limitation of these two trials was the relatively short study period (9 months), which leaves the possibility open that MRA may improve the long-term cardiovascular outcomes of patients with ESKD. Given the beneficial effects of MRAs on cardiovascular remodeling, including fibrosis, oxidative stress, and endothelial and immune function [95,96], MRA may have effects beyond just a reduction in LVM. In addition, there may have been little room for improvement associated with MRA administration because of well-preserved diastolic function and LVM at baseline and a relatively low prevalence of LVH in the participants in these trials. Taken together, the results of these two studies suggest that spironolactone at a maximum dose of 25 mg per day is likely to be reasonably safe for use in patients undergoing maintenance dialysis, provided that they are adequately monitored and any hyperkalemia is appropriately managed using a per-protocol therapeutic algorithm.

A meta-analysis performed in 2021 [97] that included the results of these trials showed that 14 RCTs had been published by 2020, 13 of which were trials of spironolactone administration and the remaining one was a trial of eplerenone administration. The sample sizes ranged from 8 to 309, the mean ages of the study samples ranged from 53 to 70 years, and the majority of the participants in most of the studies were male. The follow-up periods varied between 7 weeks and 36 months; 10 were studies of patients on hemodialysis, two were of patients on peritoneal dialysis, and two included both groups. The authors concluded that spironolactone or eplerenone could be beneficial because they showed significant reductions in cardiovascular mortality (relative risk (RR) 0.41, 95% CI 0.24–0.70, *p* = 0.001) and all-cause mortality (RR 0.44, 95% CI 0.30–0.66, *p* < 0.001) in these patients. Furthermore, there was no significant increase in the risk of hyperkalemia (RR 1.12, 95% CI 0.91–1.36, *p* = 0.29), and no significant differences were found between the groups with respect to nonfatal cardiovascular events or stroke. However, the authors noted a high risk of bias, issues with patient follow-up, and the relatively small number of studies included, which meant that the mortality data were heavily influenced by the results of just two medium-sized trials. This raises the possibility that the identified effects of MRA may have been overestimated, despite the statistically significant findings. Nonetheless, on the basis of the current evidence, MRA treatment is highly likely to be beneficial in patients undergoing dialysis. To strengthen the evidence, the detailed results of two large international major-outcome RCTs regarding the administration of spironolactone to patients undergoing chronic dialysis—the ACHIEVE (Aldosterone bloCkade for Health Improvement EValuation in End-stage Renal Disease; NCT03020303) trial and the ALCHEMIST (ALdosterone antagonist Chronic HEModialysis Interventional Survival Trial; NCT01848639)—are eagerly awaited.

## 13. Evidence for the Use of MRAs in Patients Who Have Undergone Renal Transplantation

The prevention of ischemia/reperfusion (I/R)-related adverse events and calcineurin inhibitor-induced nephrotoxicity (CIN) is important for the maintenance of kidney function and to ensure long-term graft survival in recipients of a kidney transplant [98]. Several preclinical studies have shown that the pharmacological inhibition of MR protects against I/R-related adverse events and their long-term effects [98,99]. In addition, MRAs may help prevent CIN during both the acute and chronic phases [100,101,102]. Given that many recipients have a history of CVD prior to kidney transplantation or develop these complications afterward [103], MRAs that have beneficial effects on patients with CVD may also be a promising treatment option for these patients. Although few clinical trials have focused on the use of MRAs in patients who have undergone kidney transplantation, a 6-month uncontrolled pilot study demonstrated that the addition of spironolactone (25 mg/day) to combined ACEi and ARB therapy markedly reduced proteinuria in 11 such patients (urine protein > 3 g/day and serum creatinine concentration < 265.2 μmol/L) [104]. Nine patients (81.5%) experienced a >50% reduction in their proteinuria, although their kidney function was not affected. In addition, a double-blind RCT of patients who underwent living-donor kidney transplantation showed that spironolactone significantly reduced renal oxidative stress versus placebo when administered 1 day before and 3 days after transplantation [105]. In the pediatric setting, an RCT performed on children with chronic renal allograft nephropathy showed beneficial effects of eplerenone 25 mg/day for 2 years. Eplerenone slowed the progression of the disease, according to the results of renal biopsy [106]. Although the result was not statistically significant, owing to a lack of statistical power, the eGFR of the eplerenone group was 15 mL/min/1.73 m^2^ higher than that of the control group, and the proteinuria of the eplerenone group decreased, whereas that of the control group increased.

Recently, the results of the SPIREN trial, an RCT that aimed to determine whether spironolactone would stabilize the renal function of patients who had undergone kidney transplants and were being treated with calcineurin inhibitors, have been published [107]. This trial included 188 patients, who were randomized to receive spironolactone or placebo for 3 years, in addition to their standard therapy. Spironolactone reduced the measured GFR (spironolactone −6.2 mL/min versus placebo 1.4 mL/min, 95% CI −10.9 to −4.3 mL/min), independently of the period of time elapsed since transplantation and BP, but had no effect on the kidney function curve over time. In addition, it reduced 24-h proteinuria after 1 year (spironolactone −0.09 g/day versus placebo 0.07 g/day, 95% CI −0.29 to −0.02 g/day). However, there was no difference in the eGFR slope between 6 months and 3 years, and no sustained reduction in proteinuria (after 2 or 3 years) was identified. Furthermore, spironolactone had no significant effect on the progression of renal interstitial fibrosis, assessed using renal biopsy after 2 years (spironolactone −0.52% versus placebo −3.08%, *p* = 0.47). Thus, spironolactone did not improve kidney function, reduce proteinuria in the long term, or reduce renal interstitial fibrosis over 3 years in patients who had undergone kidney transplantation, in contrast to the results of several earlier trials, which suggested that MRAs could be beneficial for such patients. Given the lack of definitive conclusions to be drawn, further evidence regarding the long-term efficacy of MRAs should be collected.

## 14. Conclusions and Perspective

MR activation increases ENaC activity in the distal nephron, resulting in greater sodium and fluid retention, which leads to hypertension. On the other hand, it also induces oxidative stress, inflammation, and fibrosis in the kidney, which accelerate the progression of CKD. MRAs alleviate heart and kidney damage by suppressing MR overactivation, thereby reducing the risk of CVD and slowing the progression of CKD. Recently, non-steroidal MRAs have been efficacious and have a superior safety profile to classical steroidal MRAs, which implies that they may become a cornerstone of the emerging treatment strategies to be used alongside existing therapies. In conclusion, MRAs offer clinicians a valuable new option for the management of hypertension and CKD. However, although the body of evidence supporting their use continues to grow, it remains insufficient. Further research is needed to identify the groups of patients that are most likely to benefit from MRA therapy and to establish optimal treatment protocols.

## Figures and Tables

**Figure 1 biomedicines-13-00053-f001:**
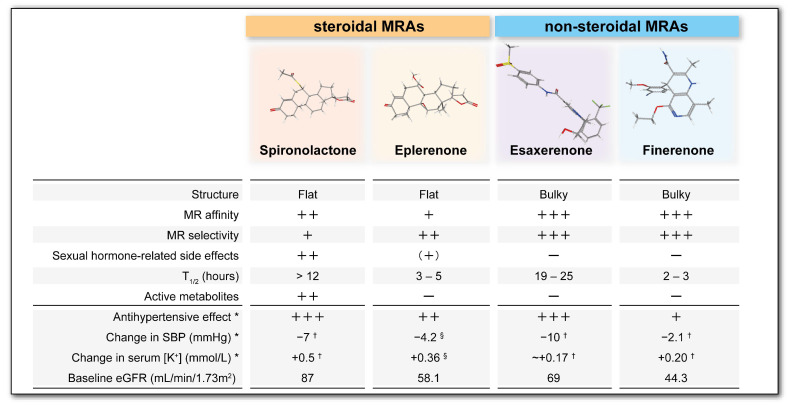
Comparison of the effects of MRAs on blood pressure and serum K^+^ concentrations. This figure shows the characteristics of the steroidal MRAs (spironolactone and eplerenone) and the non-steroidal MRAs (esaxerenone and finerenone). Finerenone has a less marked antihypertensive effect than the other three drugs, and non-steroidal MRAs appear to be less effective at increasing serum K^+^ concentrations than conventional steroidal MRAs. This figure was summarized from references [8,9,12,14,15]. The three-dimensional structure of each MRA was created using PubChem3D. *, using RAASi treatment; †, between baseline and 1 year; §, between baseline and 4 weeks.

**Figure 2 biomedicines-13-00053-f002:**
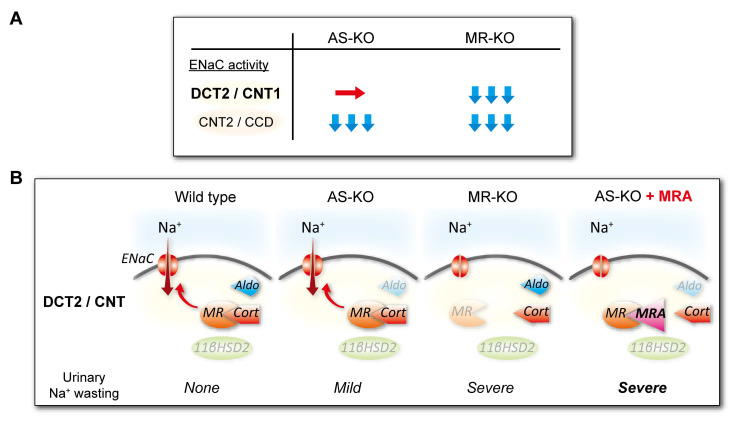
Differences in the ENaC activity and phenotype of AS-KO and MR-KO mice. (**A**) ENaC activity of AS-KO and MR-KO mice. The ENaC activity of MR-KO mice was low in both DCT2/CNT1 and CNT2/CCD, whereas that of AS-KO mice was low in CNT2/CCD but not in DCT2/CNT1. (**B**) DCT2/CNT phenotypes of wild-type, AS-KO, and MR-KO mice. Whereas MR-KO mice exhibited a salt-loss state, accompanied by a high serum K^+^ concentration and a low serum Na^+^ concentration, the AS-KO mice showed only slightly high serum K^+^ concentration and slightly low expression of phosphorylated NCC. Under conditions of LS diet-feeding, MR-KO mice showed persistently high urinary Na^+^ excretion. By contrast, the high urinary Na^+^ excretion in AS-KO mice was transient, but eplerenone treatment increased urinary Na^+^ excretion, and this was accompanied by high serum K^+^ and blood urea nitrogen concentrations, resulting in a salt-loss state similar to that of the MR-KO mice. AS: aldosterone synthase, MR: mineralocorticoid receptor, DCT2: late distal convoluted tubule, CNT1: early connecting tubule, CNT2: late connecting tubule, CCD: cortical collecting duct, Aldo: aldosterone, Cort: cortisol, 11βHSD2: 11beta-hydroxysteroid dehydrogenase type 2. These figures were based on reference [38].

**Figure 3 biomedicines-13-00053-f003:**
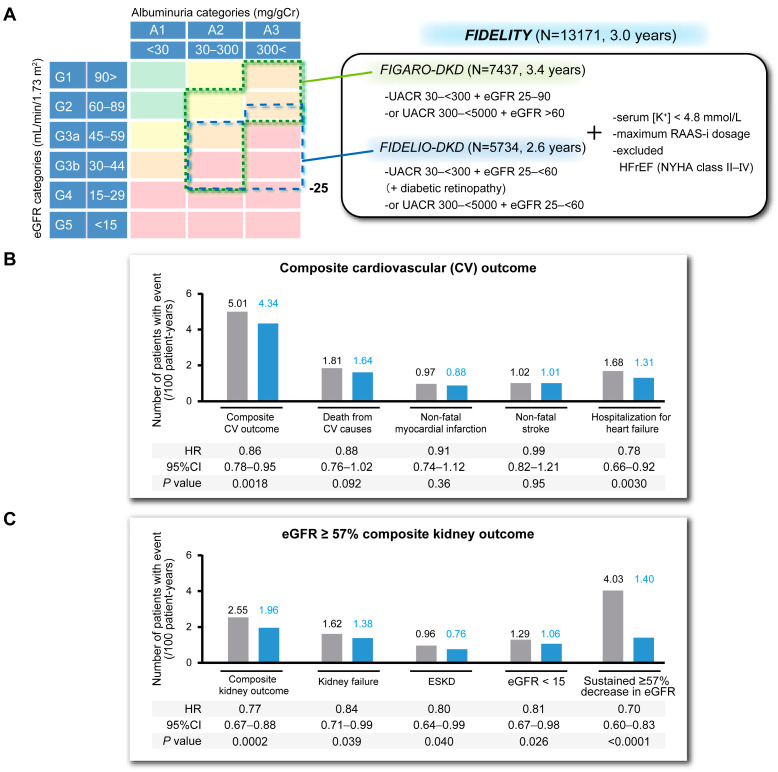
Results of the FIDELITY analysis, an analysis of data from the FIGARO-DKD and FIDELIO-DKD trials. (**A**) Inclusion and exclusion criteria for the FIGARO-DKD and FIDELIO-DKD trials. A total of 13171 patients were included in the FIDELITY integrated analysis, with a follow-up period of 3 years. (**B**) Results of the composite cardiovascular outcome in the FIDELITY analysis. Finerenone reduced the risk of an initial occurrence of the cardiovascular composite outcome by 14% and the risk of hospitalization because of heart failure by 22% versus placebo. (**C**) Results of the composite kidney outcome in the FIDELITY analysis. Finerenone reduced the risk of an initial occurrence of the composite renal outcome by 23% and the risk of the initiation of renal replacement therapy by 20% versus placebo. UACR: urinary albumin creatinine ratio, RAAS-i: renin–angiotensin–aldosterone system inhibitor, HFrEF: heart failure with reduced ejection fraction, NYHA: New York Heart Association classification, ESKD: end-stage kidney disease. These figures were summarized from references [12,13,57].

**Figure 4 biomedicines-13-00053-f004:**
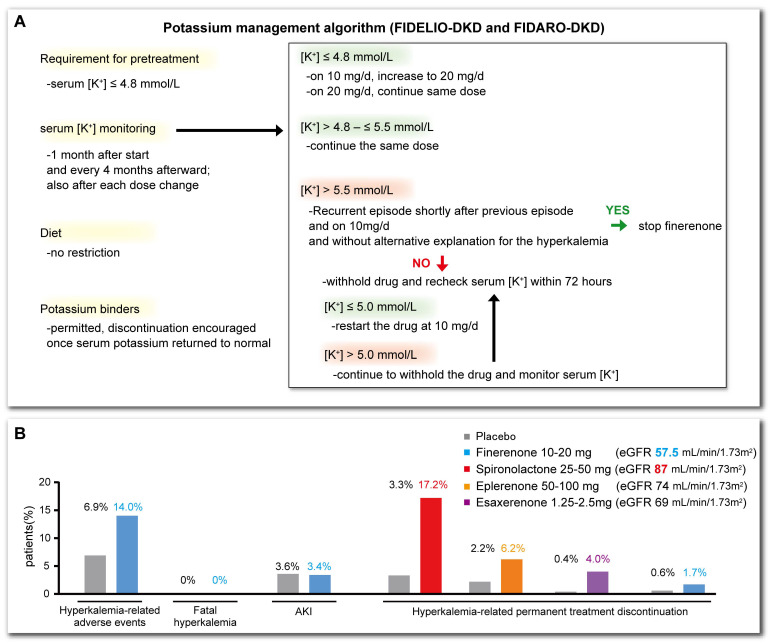
Management and incidence of hyperkalemia in the FIDELIO-DKD and FIGARO-DKD trials. (**A**) The potassium management algorithm used in the FIDELIO-DKD and FIGARO-DKD trials. (**B**) Incidences of hyperkalemia, related adverse events, and discontinuation of each MRA because of hyperkalemia. The incidence of hyperkalemia was higher in the finerenone group than in the placebo group (14.0% vs. 6.9%), but there were no deaths and few instances of severe hyperkalemia. The incidence of discontinuation owing to hyperkalemia was also lower than those for other MRAs. AKI: acute kidney injury. These figures were summarized from references [12,13,14,55,56,57].

**Figure 5 biomedicines-13-00053-f005:**
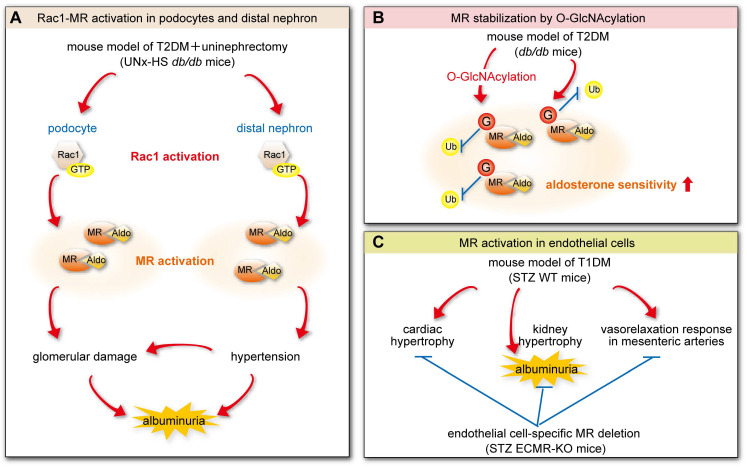
MR activation in mouse models of diabetes. (**A**) Activation of the Rac1–MR pathway in glomerular podocytes and distal nephrons. The Rac1–MR pathway was activated in *db*/*db* mice fed a high-salt diet for 10 weeks after unilateral nephrectomy versus control mice (UNx-HS *db*/*m* mice), resulting in glomerular damage and hypertension with massive albuminuria. (**B**) MR stabilization by O-GlcNAcylation. O-GlcNAcylation increased the sensitivity of the *db*/*db* mice to aldosterone by stabilizing MR activity. (**C**) MR activation in endothelial cells. The hypertrophy of the heart and kidneys, massive albuminuria, and inhibition of the vasorelaxation response in mesenteric arteries that characterized the STZ WT mice were reduced in the STZ ECMR-KO mice. T2DM: type 2 diabetes mellitus, UNx: unilateral nephrectomy, HS: high-salt, O-GlcNAcylation: O-linked-N-acetylglucosamine, T1DM: type 1 diabetes mellitus, STZ: streptozotocin, WT: wild type, ECMR: endothelial cell-specific MR. These figures were based on references [90,91,92].

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
