# Peer review of "Recent Advances and Perspectives on the Use of Mineralocorticoid Receptor Antagonists for the Treatment of Hypertension and Chronic Kidney Disease: A Review"

_biomedicines, 2024, doi:10.3390/biomedicines13010053_

Round 1

Reviewer 1 Report

Comments and Suggestions for Authors

Arounding the use of MR antagonists (MRAs) in the treatment of hypertension and chronic kidney disease (CKD), the authors of the manuscript present a detail review. The authors deserve the applaud as both the extension and the depth of the review. Besides, the authors not only present the mechanisms, the clinical, and the basic research of MRAs, but also present the limitations of MRAs in inducing side effects and the corresponding two strategies. The workload of the review is enough. Besides, the authors have provided enough insight in hypertension and CKD treatment using MRA. The review will help the understanding of MRA in hypertension and CKD therapy and the associated basic research, and help the related new drug development. Therefore, the manuscript is recommended to be accepted by the journal Biomedicines. Here are some concerns for the authors:

1) The Keywords section contain too many works. The drug name are recommended to be deleted.

2) For the Figure legends, I notice that the authors tend to use the sentence such as “This figure was adapted from references... However, the authors should notice the copyright issue. I recommend the authors to ask for the copyright and using the sentence such as: Copyright (YEAR), with permission from PUBLISHER.

3) In the 1st paragraph of section 4, the abbreviation of European Medicines Agency is recommended to be added.

4) In the 1st paragraph of section, the reference should be added to support the sentence “It undergoes hepatic metabolism and has a long half-life” instead of just citing Figure 1.

5) In the 3rd paragraph of the Introduction section, the sentence “and no pharmacologically active metabolites are generated” is recommended to be revised as “and no pharmacologically active metabolites are found.

6) In section 8, a blank space should be added (The effect of SGLT2is to reduce...3rd paragraph).

7) The section 10 talks about CKD, this section is recommended to be moved after section 8 to make the manuscript reads more logical.

Author Response

Comment 1:

1) The Keywords section contain too many works. The drug name are recommended to be deleted.

Response to comment 1:

We appreciate your important feedback. In accordance with your comment, we have deleted the drug names, except for finerenone, and have added other key words.

Comment 2:

2) For the Figure legends, I notice that the authors tend to use the sentence such as “This figure was adapted from references...” However, the authors should notice the copyright issue. I recommend the authors to ask for the copyright and using the sentence such as: “Copyright (YEAR), with permission from PUBLISHER”.

Response to comment 2:

We apologize for this lack of clarity. All Figures in this article were prepared by the authors themselves based on the references listed. To improve clarity and address any potential misunderstandings, we have revised the corresponding texts from “adapted from” to “summarized from” or “based on”, as appropriate.

Comment 3:

3) In the 1st paragraph of section 4, the abbreviation of European Medicines Agency is recommended to be added.

Response to comment 3:

We appreciate your comment. We have revised it as you pointed out.

Comment 4:

4) In the 1st paragraph of section, the reference should be added to support the sentence “It undergoes hepatic metabolism and has a long half-life” instead of just citing Figure 1.

Response to comment 4:

Thank you for your valuable suggestions. We have added the references 8 and 9 at the end of the relevant sentence.

Comment 5:

5) In the 3rd paragraph of the Introduction section, the sentence “and no pharmacologically active metabolites are generated” is recommended to be revised as “and no pharmacologically active metabolites are found”.

Response to comment 5:

We appreciate your comment. We have revised it as suggested.

Comment 6:

6) In section 8, a blank space should be added (The effect of SGLT2is to reduce...3rd paragraph).

Response to comment 6:

Thank you for your suggestions. As defined earlier, SGLT2is refers to SGLT2 inhibitors, so we believe a blank space is unnecessary.

Comment 7:

7) The section 10 talks about CKD, this section is recommended to be moved after section 8 to make the manuscript reads more logical.

Response to comment 7:

Thank you for your valuable suggestions. In accordance with your advice, we have moved Section 10 to follow Section 8, ensuring a more logical flow in the manuscript. Additionally, we have updated the numbering of the relevant references. The text in the legend of Figure 5 has been revised to reflect these adjustments.

Reviewer 2 Report

Comments and Suggestions for Authors

Very actual subject with a lot of practical implications both for cardiologists and nephrologists.

The article is well conceived, very well structured.

there are some things I have to mention:

At references: why numbers are written two times?

Ref no 1,2  very old, perhaps for historical reasons.

References are not uniformly written – some of them with too many authors.

At ref 15 I think not all the authors are necessary to be mentioned. 

Author Response

Response to Reviewer #2

Comment 1:

At references: why numbers are written two times?

Response to comment 1:

We apologize for any confusion and appreciate your appropriate advice. As recommended in the “Instructions for Authors” section of the Biomedicines website, we used EndNote for the preparation of the references. In response to your comments, we have carefully re-checked all references to confirm their accuracy and consistency with Pubmed. We found no discrepancies, and no changes were necessary.

Comment 2:

Ref no 1,2 very old, perhaps for historical reasons.

Response to comment 2:

We appreciate your insightful advice. Although references 1 and 2 are dated, we believe those are the original papers and are essential for illustrating the historical progression of research on mineralocorticoid receptors.

Comment 3:

References are not uniformly written – some of them with too many authors.

At ref 15 I think not all the authors are necessary to be mentioned. 

Response to comment 3:

We appreciate your comment. We have revised the text accordingly. In addition, references 6 and 49 have also been shortened, since they contained an extensive number of authors.

Reviewer 3 Report

Comments and Suggestions for Authors

Congratulation for this article,it is a hard work behind.The thema is of interest for nephrologists and cardiologists.I have only minor comments:

-row 138 "more effective at......blockers". Please be more precise,what do you mean by more effective?It is advisable to provide a value of a statistical test.

-in general,also a narrative review must follow the steps:Introduction,Methods,Results and Conclusions.I advise to insert a Method part,in which to explain briefly how did you selected the articles.

Author Response

Comment 1:

-row 138 "more effective at......blockers". Please be more precise, what do you mean by more effective? It is advisable to provide a value of a statistical test.

Response to comment 1:

We appreciate your insightful comment. We have modified the text to include the predicted changes in blood pressure at a plasma renin level of 1 mU/L to make this clear. However, since no specific values were provided in the text or in the supplementary appendix, we have estimated the values for doxazosin and bisoprolol based on the graphs.

Comment 2:

-in general, also a narrative review must follow the steps: Introduction, Methods, Results and Conclusions. I advise to insert a Method part, in which to explain briefly how did you selected the articles.

Response to comment 2:

Thank you for your valuable comment. We have created a section on methods and added it as Section 2.